# The Novel Cardiac Myosin Activator Danicamtiv Improves Cardiac Systolic Function at the Expense of Diastolic Dysfunction In Vitro and In Vivo: Implications for Clinical Applications

**DOI:** 10.3390/ijms24010446

**Published:** 2022-12-27

**Authors:** Arnold Péter Ráduly, Fruzsina Sárkány, Máté Balázs Kovács, Brigitta Bernát, Béla Juhász, Zoltán Szilvássy, Róbert Porszász, Balázs Horváth, Norbert Szentandrássy, Péter Nánási, Zoltán Csanádi, István Édes, Attila Tóth, Zoltán Papp, Dániel Priksz, Attila Borbély

**Affiliations:** 1Division of Clinical Physiology, Department of Cardiology, Faculty of Medicine, University of Debrecen, 4032 Debrecen, Hungary; 2Division of Cardiology, Department of Cardiology, Faculty of Medicine, University of Debrecen, 4032 Debrecen, Hungary; 3Kálmán Laki Doctoral School, University of Debrecen, 4032 Debrecen, Hungary; 4Department of Pharmacology and Pharmacotherapy, Faculty of Medicine, University of Debrecen, 4032 Debrecen, Hungary; 5Department of Physiology, Faculty of Medicine, University of Debrecen, 4032 Debrecen, Hungary; 6Department of Basic Medical Sciences, Faculty of Dentistry, University of Debrecen, 4032 Debrecen, Hungary; 7Department of Dental Physiology and Pharmacology, Faculty of Dentistry, University of Debrecen, 4032 Debrecen, Hungary; 8HAS-UD Vascular Biology and Myocardial Pathophysiology Research Group, Hungarian Academy of Sciences, 4032 Debrecen, Hungary

**Keywords:** heart failure with reduced ejection fraction, myosin activators, danicamtiv, positive inotropy, strain, diastolic dysfunction

## Abstract

Recent cardiotropic drug developments have focused on cardiac myofilaments. Danicamtiv, the second direct myosin activator, has achieved encouraging results in preclinical and clinical studies, thus implicating its potential applicability in the treatment of heart failure with reduced ejection fraction (HFrEF). Here, we analyzed the inotropic effects of danicamtiv in detail. To this end, changes in sarcomere length and intracellular Ca^2+^ levels were monitored in parallel, in enzymatically isolated canine cardiomyocytes, and detailed echocardiographic examinations were performed in anesthetized rats in the absence or presence of danicamtiv. The systolic and diastolic sarcomere lengths decreased; contraction and relaxation kinetics slowed down with increasing danicamtiv concentrations without changes in intracellular Ca^2+^ transients in vitro. Danicamtiv evoked remarkable increases in left ventricular ejection fraction and fractional shortening, also reflected by changes in systolic strain. Nevertheless, the systolic ejection time was significantly prolonged, the ratio of diastolic to systolic duration was reduced, and signs of diastolic dysfunction were also observed upon danicamtiv treatment in vivo. Taken together, danicamtiv improves cardiac systolic function, but it can also limit diastolic performance, especially at high drug concentrations.

## 1. Introduction

Deteriorating left ventricular (LV) systolic performance is thought to be responsible for the high morbidity and mortality of patients suffering from heart failure with reduced ejection fraction (HFrEF) [1,2]. Based on clinical evidence, pharmacological therapies of chronic HFrEF primarily aim to interfere with neuro-humoral signaling via the renin–angiotensin–aldosterone system (RAAS), the sympathetic nervous system (SNS), and the natriuretic peptide system (NPS) [3]. However, these medications have no direct effects on the contractility of LV cardiomyocytes. Conversely, positive inotropic (PI) agents do increase systolic performance either by augmenting the intracellular Ca^2+^ availability (e.g., beta-receptor agonists, PDE III inhibitors, Na^+^/Ca^2+^-exchanger modulators) or by fine-tuning the Ca^2+^ responsiveness of the contractile protein machinery (e.g., Ca^2+^-sensitizers) of cardiomyocytes, though with uncertain effects on the long-term prognosis of HFrEF.

A new class of positive inotropic drugs, the myosin activators, gained considerable interest about a decade ago. These agents modulate the function of the myosin motor protein by acting on the chemo-mechanical actin–myosin crossbridge cycle [4,5]. Importantly, myosin activators augment cardiac contractions without increasing the oxygen demand of the myocardium and arrhythmia incidence [6]. The first representative of these myotropes is omecamtiv mecarbil (OM), which achieved promising results in clinical trials in association with increases in LV ejection fraction (EF), systolic ejection time (SET), and with no major side effects [7]. Moreover, a large-scale phase III clinical trial with OM has recently revealed a lower incidence of a composite endpoint of heart failure (HF) event or death from cardiovascular (CV) causes in patients with chronic HFrEF [8]. Nevertheless, potential increases in the level of circulating troponins stemming from LV diastolic dysfunction, a narrow therapeutic window, and undesired side effects on skeletal muscles raised concerns for the clinical applicability of OM in HFrEF [9,10,11].

Recent pharmacological efforts led to the development of a new direct myosin activator, danicamtiv, which also binds to the myosin molecule of cardiomyocytes. The dani-camtiv binding site on myosin, and consequently its molecular action, is thought to differ from those of OM [5,12]. Moreover, the selectivity of danicamtiv for cardiac myosin over skeletal or smooth muscle myosin isoforms is considered to be higher than for OM [13]. Nevertheless, similarly to OM, danicamtiv also accelerates the ATPase turnover rate of purified human S1 myosin molecules in vitro, and increases SET and LVEF with additional improvements in the left atrial systolic function in an experimental dog model of HFrEF in vivo [13]. These findings prompted a dose-finding trial in humans where, besides signs of improved LV systolic function, transient asymptomatic increases in troponin levels were also detected in 23% of the patients involved [13]. Nevertheless, the results of a phase 1 clinical trial and a pharmacokinetic study also supported a good safety and tolerability profile for danicamtiv in human subjects [14,15,16]. In comparison to OM administrations, a wider therapeutic window, a greater potential to improve contractile function, and at lower lusitropic costs, were associated with the applications of danicamtiv in a recent preclinical study conducted in human engineered heart tissues [17], although it should be also noted that the preclinical and clinical data on danicamtiv are still scarce and to some degree obscure, and in particular for the diastolic effects of this new myotrope [13,17,18].

In the present study, we aimed to address the contractile effects of danicamtiv at the sarcomere and organ levels. Here, we show the results of a detailed concentration-response analysis in isolated canine cardiomyocytes where danicamtiv effects on sarcomere dynamics are interpreted in view of intracellular Ca^2+^ concentration changes. Complimentary state-of-the-art echocardiographic examinations in rat hearts allowed the analysis of danicamtiv’s effects on LV contractile performance. The combination of in vitro and in vivo studies served as a tool to elucidate the relationship between danicamtiv’s evoked effects on myocardial systolic and diastolic functions. Our results illustrate danicamtiv as a direct myosin activator having a largely similar pharmacological profile to OM and signify the potential for diastolic dysfunction during direct myosin activation in general.

## 2. Results

### 2.1. Danicamtiv Altered LV Dimensions and Cardiomyocyte Mechanics

The characteristics of danicamtiv’s evoked changes in myocardial function were monitored by echocardiography in anaesthetized rats (Figure 1A–D) and by cellular physiological methods in enzymatically isolated canine cardiomyocytes (Figure 1E–H). A significant decrease in the LV end-systolic diameter (ESD), but no change in the LV end-diastolic diameter (EDD) were observed upon the intravenous administration of 2 mg/kg danicamtiv (representative examples shown in Figure 1A,B, EDD: from 7.56 ± 0.12 to 7.74 ± 0.26 mm, *p* = 0.45, Figure 1C; ESD: from 3.76 ± 0.11 to 2.93 ± 0.17 mm, *p* < 0.001, Figure 1D) in rats. Surprisingly, in isolated cardiomyocytes a concentration-dependent decrease in diastolic sarcomere length (SL) was observed as illustrated in Figure 1E upon danicamtiv administration during steady-state field stimulation at a frequency of 0.5 Hz at room temperature (i.e., from 1.91 ± 0.01 (Control) to 1.90 ± 0.01, 1.89 ± 0.01, 1.86 ± 0.01, 1.83 ± 0.01, 1.69 ± 0.03, and 1.57 ± 0.04 µm at 0.01, 0.1, 0.3, 0.5, 1 or 2 µM danicamtiv, respectively; significant for SL changes at danicamtiv concentrations 0.5, 1, and 2 µM, P = 0.002, *p* < 0.001, *p* < 0.001, respectively) (Figure 1G). Danicamtiv at concentrations of 1 and 2 µM also evoked significant reductions in systolic SL (from 1.65 ± 0.02 (Control) to 1.51 ± 0.02 and 1.45 ± 0.03 µm, *p* = 0.003 and *p* < 0.001, respectively, Figure 1H). The decreases in diastolic and systolic SL were accompanied by an increase in the durations of cardiomyocyte contractions (Figure 1E). The time courses of the Ca^2+^ transients were not affected by danicamtiv treatments (Figure 1F).

### 2.2. Danicamtiv Improved Left Ventricular Contractile Function In Vivo

The relevant LV systolic contractile parameters, including LVEF (from 79.76 ± 1.24 to 89.36 ± 0.99%, *p* < 0.001), fractional shortening (FS) (from 50.58 ± 1.30 to 63.57 ± 1.47%, *p* < 0.001), and stroke volume (SV) (from 243.89 ± 10.07 to 290.56 ± 20.26 µL, *p* = 0.012) increased significantly upon the intravenous administration of 2 mg/kg danicamtiv (Figure 2A–C). In addition, cardiac output (CO) also increased (from 61.40 ± 2.99 to 72.45 ± 5.04 mL/min, *p* = 0.026) (Figure 2D), supporting the positive inotropic effect of the drug, and the consequent hemodynamic improvement.

### 2.3. Danicamtiv Increased Contraction Durations in Rats and Canine Cardiomyocytes

Both systolic durations (from 88.64 ± 1.83 to 106.00 ± 3.30 ms, *p* = 0.0001) and systolic ejection time (SET) were significantly prolonged upon danicamtiv treatments in rats (from 72.28 ± 1.53 to 87.14 ± 3.05 ms, *p* < 0.001) (Figure 3A,B). Similarly, a progressive prolongation in contraction duration (Figure 3C) and in the time to peak interval (Figure 3D) were observed upon increasing danicamtiv concentrations in canine cardiomyocytes (0.64 ± 0.03, 0.78 ± 0.06, 0.94 ± 0.06, 1.02 ± 0.06, 1.45 ± 0.08 and 2.06 ± 0.22 s, at 0.01, 0.1, 0.3, 0.5, 1, and 2 µM danicamtiv concentrations, vs. 0.66 ± 0.03 s (Control), respectively, *p* < 0.001 for drug concentrations ≥ 0.3 µM for contraction duration; and 0.49 ± 0.02, 0.69 ± 0.09, 1.00 ± 0.10, 1.19 ± 0.08, 1.04 ± 0.06, and 1.57 ± 0.21 s, at 0.01, 0.1, 0.3, 0.5, 1, and 2 µM danicamtiv concentrations vs. 0.51 ± 0.02 s (Control), respectively, *p* = 0.029, *p* = 0.004, *p* < 0.001 and *p* < 0.001 for 0.3, 0.5, 1, and 2 µM drug concentrations for time to peak interval). Echocardiography revealed a significant decrease in the duration of diastole (from 152.20 ± 4.19 to 133.00 ± 4.66 ms, *p* = 0.008) (Figure 3E), which was also evidenced by a marked decrease in the ratio of diastolic to systolic durations (from 1.73 ± 0.06 to 1.28 ± 0.07, *p* < 0.001, Figure 3F).

### 2.4. Danicamtiv Decreased Myocardial Contraction Kinetics Both In Vivo and In Vitro

When danicamtiv was applied to rats, a significant decrease in LV peak radial systolic velocity (from 3.50 ± 0.13 to 2.67 ± 0.17 cm/s, *p* < 0.001, N = 9, Figure 4A) and contraction velocity (M-mode slope) (from 3.32 ± 0.23 to 2.30 ± 0.11 cm/s, *p* = 0.008, N = 10 Figure 4B) were observed. The lengthening of LV contractions could be attributed to the slower kinetics of cardiomyocyte contraction (from 1.16 ± 0.10 to 1.02 ± 0.12, 0.85 ± 0.09, 0.76 ± 0.08, 0.68 ± 0.08, 0.39 ± 0.07, and 0.18 ± 0.03 µm/s, respectively) with increasing danicamtiv concentrations (control, 0.01, 0.1, 0.3, 0.5, 1, and 2 µM, respectively) significant at danicamtiv concentrations 0.5, 1, and 2 µM, *p* = 0.021, *p* < 0.001 and *p* < 0.001 (Figure 4C). Upon danicamtiv administration, both global longitudinal strain (from −24.22 ± 1.30 to −33.27 ± 1.45%, *p* < 0.001, Figure 4D) and global circumferential strain (from −54.58 ± 3.28 to 59.29 ± 3.36%, *p* = 0.015, Figure 4E) decreased significantly, thus indicating LV systolic function improvement. In addition, a marked increase in the pulmonary vein atrial reversal flow velocity (from 136.70 ± 11.35 to 201.10 ± 24.85 mm/s, *p* = 0.008) and atrial reversal flow duration (from 20.27 ± 0.77 to 28.10 ± 2.24 ms, *p* = 0.004) were observed, which indicate an augmentation of the left atrial systolic function. No significant changes were seen in the aortic or pulmonary artery mean and peak velocities or gradients developed during danicamtiv treatments (Table 1).

### 2.5. Danicamtiv Impaired Diastolic Function

Danicamtiv’s administration was associated with a significant decrease in the early (E) transmitral filling velocity (from 720.92 *±* 11.87 to 645.92 *±* 29.17 mm/s, *p* = 0.013) and an increase in the late (A) transmitral filling velocity (from 379.24 *±* 19.00 to 506.93 *±* 37.81 mm/s, *p* = 0.003, Figure 5A–D). The consequent decrease in the E/A ratio (from 1.97 *±* 0.10 to 1.38 *±* 0.10, *p* = 0.0003, Figure 5E) indicated an elevation in left ventricular filling pressures. The tissue doppler imaging (TDI) velocities (i.e., mitral valve septal early (a’) and late (e’) filling tissue velocities) tended to change similarly to the mitral E and A wave velocities (Table 1). The changes in the TDI e’/a’ ratio further support the altered mitral inflow pattern upon danicamtiv administration. Nevertheless, the isovolumic relaxation time (IVRT) was not altered by danicamtiv. Of note, the alterations in the indices of ventricular diastolic function were mirrored by the slower relaxation velocities in isolated cardiomyocytes (i.e., from the drug-free control of 1.63 *±* 0.15 µm/s, to 1.48 *±* 0.20, 1.39 *±* 0.13, 1.13 *±* 0.12, 0.98 *±* 0.11, 0.48 *±* 0.10, and 0.20 *±* 0.05 µm/s, with increasing danicamtiv concentrations; 0.01, 0.1, 0.3, 0.5, 1, and 2 µM, respectively, *p* < 0.001 at danicamtiv concentrations *≥* 1 µM Figure 5G).

### 2.6. Danicamtiv Did Not Affect the Time Courses of Ca^2+^ Transients and Had No Proarrhytmic Effects

During the cumulative administration of danicamtiv (0.01, 0.1, 0.3, 0.5, 1, and 2 µM) no changes in the amplitudes and durations of Ca^2+^ transients were observed (Figure 6A,B). Moreover, no changes could be detected when the kinetic parameters of Ca^2+^ transients were studied (data not shown). When electrocardiac parameters were analyzed (heart rate, PQ, QRS duration, QT interval, and T wave amplitude), no relevant changes or arrhythmias were observed upon danicamtiv application (Table 1).

## 3. Discussion

The results of our present investigations illustrate danicamtiv, a new direct myosin activator with an apparent coupling between its systolic and diastolic effects. Our data extend previous danicamtiv-related observations and shed new light on its characteristics for a potential future clinical application as a positive inotrope.

Targeted disease-modifying therapies appear as emerging new strategies in current HF guidelines to treat cardiomyopathies [3]. The myosin inhibitor mavacamten by interfering with the hypercontractile state of cardiomyocytes may also temper remodeling in hypertrophic cardiomyopathy [19,20]. However, there is no specific therapy for dilated cardiomyopathy (DCM) to date. This fact and the experience with OM have encouraged the development of the second myosin activator, with danicamtiv reaching the clinical stage [19,21].

The binding site of direct myosin activators is located on the S1 domain of the β-myosin heavy chain in the cardiomyocytes of canine and rodent hearts. Of note, danicamtiv (similarly to OM) had no effect on intracellular Ca^2+^ transients in cardiomyocytes, suggesting that the above drug–target interaction is solely responsible for its cardiotonic effects. It follows that danicamtiv-evoked contractile responses will be largely comparable in dogs and rats, as has been formerly also evidenced for OM in identical canine and rodent cardiac preparations [22]. Here, we decided to conduct our cellular physiological experiments in canine cardiomyocytes considering their electrophysiological similarity to human cardiomyocytes [23], and to complement our investigations at the organ level in rats as current echocardiographic technologies allow high-quality imaging in small mammals [24].

The pathophysiological processes occurring in HFrEF lead to significant declines in contractility with manifest reductions of SV [25]. Positive inotropic agents have been considered for the compensation of this cardiac malfunction both in acute and chronic HFrEF for several decades [4,5,22]. Conventional calcitropes increase intracellular Ca^2+^ levels, oxygen consumption, and also mortality in the long term [26,27,28]. In contrast, direct myosin activators exert an allosteric effect on the myosin ATPase, thus allowing more actin-myosin contacts to be formed for an unchanged intracellular Ca^2+^ transient, a mechanism that supposedly avoids the above side effects.

Here, we performed comprehensive experiments and showed that danicamtiv has major effects on both the systolic and diastolic functions of enzymatically isolated intact canine LV cardiomyocytes in vitro and in rat hearts in vivo. Danicamtiv evoked a significant decrease both in systolic and diastolic SL in our in vitro experiments, similarly to OM [10,29]. The reduction in diastolic SL could be associated with an increase in cardiomyocyte passive tension, measured at low (diastolic) Ca^2+^ levels, which has been also demonstrated during OM administration [10]. Of note, an increase in passive tension served as an explanation for LV diastolic dysfunction in patients with HF with preserved EF (HFpEF) [30]. The above data on OM and the results of a preclinical investigation on cardiac S1 myosin with omecamtiv and danicamtiv are suggestive of an increase in the myosin ATPase rate even at diastolic Ca^2+^ levels by direct myosin activators [11,20].

Our current assessment on LV contractility revealed significant increases in FS, SV, EF, and CO upon danicamtiv administration, similarly to previous findings obtained with OM [7]. In contrast to OM, which decreased both EDD and ESD in humans [7], here we found that danicamtiv decreased only ESD but not EDD in rat hearts. These findings are in accordance with the preclinical data of an experimental canine model of HF, and to some degree also with the results of a clinical phase 2a trial, where EDD appeared somewhat less sensitive for danicamtiv than ESD [13]. In a recent study, where the effects of danicamtiv and OM were compared in a human engineered myocardium, danicamtiv led also to a greater augmentation of systolic contraction with less negative effects on relaxation [17]. Increased LV filling pressure is known to be compensated by an increase in atrial contractile function. In the case of danicamtiv, this phenomenon was well reflected by a significant increase in the transmitral A wave and a decrease in the E wave; thus, a characteristic alteration in the E/A ratio, a major parameter of diastolic function. Accordingly, these changes might be sufficient to compensate for an adequate LV filling. Taken together, our data and that of others implicate higher gains in SV and CO and more favorable diastolic effects for danicamtiv than for OM. Moreover, we propose that the apparent discrepancy between the danicamtiv-evoked decrease in the diastolic SL of unloaded cardiomyocytes and the unchanged EDDs is the consequence of danicamtiv-aided atrial compensation during LV filling.

The in vivo and in vitro results with danicamtiv showed similarities when comparing the kinetic parameters of the contraction and relaxation of canine cardiomyocytes to those of systolic and diastolic durations in rat LVs: i.e., contraction kinetics was significantly slowed down in vivo, while time to peak and contraction time values were significantly increased in vitro. The increase in systolic ejection time and systolic contraction time appear as hallmarks of myosin activators, as was shown first for OM treatments [11,22]. Of note, an increased systolic time results in a shortened diastolic duration, so that the ratio of diastolic to systolic durations will also be affected by danicamtiv treatment, with potential implications for coronary perfusion and troponin release. In contrast to the effects of OM in humans [7], the isovolumic contraction time (IVCT) and IVRT were not affected by danicamtiv in rat hearts. Nevertheless, SET was significantly increased by danicamtiv, suggestive for an improvement in the myocardial performance index (Tei index) by danicamtiv.

Novel echocardiographic parameters, such as peak radial systolic velocity from strain analysis and the conventional M-mode slope, were also consistent with kinetic parameters measured in canine cardiomyocytes in vitro. Here, we evaluated myocardial strain and velocity by using speckle tracking (STE, non-Doppler, or 2-dimensional strain) echocardiography. Strain rate imaging is a novel approach for assessing myocardial function and has not been included in routine diagnostic methods yet. It is important to note that strain echocardiography, especially in small animals, has also been also less studied, although it provides a very sensitive approach to detect alterations in LV function. Myocardial velocity and strain can be also measured by TDI, though STE 2D-strain is considered to be a superior technique since, in contrast to TDI, the obtained results are angle-independent [31]. In general, data obtained by 2D-strain show a good correlation with sonomicrometry and magnetic resonance imaging (MRI), the gold-standard tool of myocardial deformation and volume analysis [31]. Here, danicamtiv treatment was shown to decrease the contraction velocity of the myocardium, which was also confirmed by a significant decrease in a 2D-strain parameter, the peak radial systolic velocity.

Additionally, 2D-strain echocardiography seems to be a more sensitive tool to assess LV function than measuring EF by classical methods. For example, it has been reported that global longitudinal strain or circumferential strain (GLS or GCS, respectively) can convincingly detect systolic dysfunction even when EF is normal [32]. In our present study, the effects of danicamtiv on LV systolic function were evaluated not only by standard methods (measuring EF, FS, SV, CO, and aortic velocities) but also by 2D-strain echocardiography. We found significantly decreased GLS and GCS values in danicamtiv-treated rat hearts, further confirming the positive effects of the drug on LV contractility as was also reported for danicamtiv in a phase 2a trial in HFrEF patients and in a dog model of HFrEF [13].

It is to be acknowledged that extrapolation of our results to the failing heart could be limited by the fact that our investigations were carried out on the cardiac preparations of healthy animals. Nevertheless, our results are in accord with those performed in a canine model of heart failure and in a phase 2a trial on danicamtiv [13].

Taken together, the cardiac effects of OM and danicamtiv are largely similar: both compounds effectively enhance LV systolic function, although they can also limit LV diastolic function. The data presented here implicate advantages for danicamtiv over OM, since danicamtiv did not affect IVRT or IVCT. Moreover, left atrial contractile function was enhanced at a relatively low danicamtiv concentration. To our knowledge, similar effects have not been reported for OM yet.

## 4. Materials and Methods

### 4.1. Animals

#### 4.1.1. In Vitro Experiments

Cardiomyocytes with intact membranes isolated from adult mongrel dogs (N = 6 weighed 10.00 ± 3.03 kg (6 to 14 kg), aged 15.03 ± 1.02 months (13.9 to 16.9 months)) were anesthetized with intramuscular injections of ketamine hydrochloride (10 mg/kg; Calypsol, Richter Gedeon, Hungary) and xylazine hydrochloride (1 mg/kg; Sedaxylan, Eurovet Animal Health BV, The Netherlands) according to a protocol approved by the local Animal Care Committee (2/2020/DEMÁB).

#### 4.1.2. In Vivo Experiments

In vivo experiments were performed on 8–12-week-old adult Sprague-Dawley rats (417.90 ± 51.95 g, Charles River Laboratories Inc., Wilmington, Germany) and they were housed in a room with controlled temperature and kept under 12/12 h dark/light cycle. Rats were anaesthetized with ketamine/xylazine combination (75/5 mg/kg, respectively); thereafter, chest hair was shaved, tail vein was canulated (the cannula was removed at the end of the experiments), and animals were positioned in a dorsal position on a heating pad (39 °C). During echocardiographic examinations, external 3-lead ECG registration was continuously performed. Danicamtiv was administered intravenously at a dose of 2 mg/kg according to a protocol approved by the local Animal Care Committee (4-1/2019/DEMÁB).

### 4.2. Drugs and Chemicals

Chemicals were obtained from Sigma-Aldrich Co. (St. Louis, MO, USA). Danicamtiv was purchased from MedChemExpress (Monmouth Junction, NJ, USA). Stock solutions for in vitro experiments were prepared in DMSO as solvent and stored at 4 °C. Appropriate amounts of concentrated stock solutions were dissolved in the bathing medium to obtain final danicamtiv concentrations of 0.01, 0.1, 0.3, 0.5, 1, and 2 μM for in vitro experiments. These concentrations were reached by cumulative dosing during the experiments. The concentration of DMSO in these solutions was 0.1%. The control (danicamtiv-free) vehicle contained the same amount of DMSO. With regard to in vivo experiments, danicamtiv was administered intravenously (*iv*.), dissolved in its special solvent (10% DMSO/90% (20% SBE-β-CD in saline)) at a dose of 2 mg/kg. Danicamtiv dose was chosen according to a previously published pharmacokinetic study [16]. Control experiments were performed with solutions containing only the vehicle (10% DMSO/90% (20% SBE-β-CD in saline)).

### 4.3. Isolation of Canine Left Ventricular Cardiomyocytes

Canine cardiomyocytes obtained from adult mongrel dogs were studied, since their physiological properties are very similar to those of humans. Cardiomyocytes were isolated from the midmyocardial region of the LV, as was described previously [33,34]. Briefly, hearts were isolated from anesthetized (ketamine-HCl 10 mg/kg, xylazine-HCl 1 mg/kg) adult mongrel dogs. Thereafter, the hearts were applied to a perfusion system and the left anterior descending coronary artery was cannulated. Single cardiomyocytes were obtained by enzymatic dispersion technique by first using Ca^2+^ free Joklik solution (Minimum Essential Medium Eagle, Joklik Modification; Sigma-Aldrich Co., St. Louis, MO, USA) for 5 min, followed by 30 min-long perfusions with Joklik solution containing 1 mg/mL collagenase (Type II, Worthington Biochemical Co., Lakewood, NJ, USA) and 0.2% bovine serum albumin (Fraction V, Sigma) added 50 µM Ca^2+^. Then cells were grounded, and normal Ca^2+^ concentration was restored progressively. Cells were stored at 15 °C until the measurements.

### 4.4. Recording of Intracellular Ca^2+^ Transients and Cardiomyocyte Shortening

Cardiomyocytes were loaded with 5 μM Fura-2 AM Ca^2+^-sensitive ratiometric fluorescent dye for 30 min in the presence of Pluronic F-127 (25 mg/mL) to avoid early elimination of the dye from the intracellular space. A total of 25 mg Pluronic F-127 was dissolved in 1 mL DMSO and this solvent was used to make a Fura-2 AM stock solution. Cells were then incubated for 30 min to allow the intracellular esterases to release Fura-2. Cardiomyocytes were placed in a chamber on the stage of an inverted microscope (Nikon TS-100). The final volume of the chamber was filled with 1 mL Tyrode solution (containing 144 mM NaCl, 5.6 mM KCl, 2.5 mM CaCl_2_, 1.2 mM MgCl_2_, 5 mM HEPES, and 11 mM dextrose, pH = 7.4). After sedimentation, a rod-shaped cardiomyocyte with clear striation, and acceptable contraction upon field-stimulation was selected for further experiments. Field stimulation was performed at 0.5 Hz (Experimenta Setup, MDE, Heidelberg). Alternating excitation wavelengths of 340 nm and 380 nm were used to monitor the fluorescence signals of Ca^2+^-bound and Ca^2+^-free Fura-2 dye, respectively. Fluorescent emission was detected above 510 nm in the case of both wavelengths, and traces were digitized at 120 Hz using the FeliX Software (Ratiomaster RM-50 system, Horiba, New Brunswick, NJ, USA) [35].

### 4.5. Determination of Effects on Diastolic SL, Contractile Parameters, and Ca^2+^ Transients

The experimental protocol was the following: cardiomyocytes were paced at 0.5 Hz for at least 2–3 min to achieve a steady state at the beginning of each experiment. Da-nicamtiv was added in a cumulative manner (final concentration of 0.01, 0.1, 0.3, 0.5, 1 or 2 µM) followed by a 4–7-min incubation period, then contractility was measured upon 0.5 Hz pacing. Multiple parameters of cardiomyocyte contraction and intracellular Ca^2+^ transients were assessed. Cardiomyocyte SL was measured in µm by a high-speed camera. Duration of contraction was defined as the time in seconds from the beginning until the end of cardiomyocyte contractions and time to peak interval was assessed from the beginning to the peak of shortenings, also in seconds. Rates of contraction and relaxation were determined at the linear phases of contractions and relaxations, respectively, and expressed in µm/s. The resting Ca^2+^ level was estimated by Fura-2 ratio (fluorescent intensity ratio at 340 and 380 nm excitation) at baseline (before cardiomyocyte stimulation). The Ca^2+^ transient duration was determined as the time interval measured between the 50% of ascending and descending slope of Ca^2+^ transient. The amplitude of Ca^2+^ transients was defined as the difference between the peak of Ca^2+^ transients and the resting Ca^2+^ levels (340 nm/380 nm ratio). Ca^2+^ transient increase kinetics was determined as the slope of a linear regression line fitted to the ascending Ca^2+^ transient (340/380*s−1).

### 4.6. Echocardiography

Echocardiography was carried out using the Vevo 3100 Imaging System including Vevo Imaging Station (VisualSonics, Amsterdam, The Netherlands) equipped with high-frequency transducer (MX250, 14–28 MHz). Anesthetized rats (ketamine-HCl 10 mg/kg, xylazine-HCl 1 mg/kg) were shaved and placed onto a heated platform (VisualSonics SR200) equipped with ultra-low noise, high-resolution ECG electronics. The ECG was continuously monitored and recorded (2 min) during echocardiographic examinations. Echocardiographic imaging was started 5 min after danicamtiv i.v. injection and lasted maximum 15 min. Data acquisition was performed in B-, 2D-, M-, and Doppler modes, from parasternal long- and short axis (PSLAX, PSAX, respectively), as well as suprasternal and apical 4 chamber views (SST, A4C, respectively). Cardiac function was assessed in accordance with the guidelines of American Society of Echocardiography [36]. The manufacturer’s recommendations for focus, 2D gain, picture width, and depth were used to optimize B-Mode imaging quality. Heart rate (HR, bpm) was automatically calculated from the ECG R-R interval data of 5 s. Wall thickness and chamber diameters were measured in M-mode, at mid-level of the papillary muscles, from both PSLAX and PSAX views, left atrial (LA) maximal diameter (mm), aortic root (Ao) diameter (mm) from M-mode recordings. End diastolic diameter (EDD, mm) and end systolic diameter (ESD, mm), as well as anterior and posterior LV wall thickness (mm, LVAWd, LVAWs, and LVPWd, LVPWs, respectively) and interventricular septum thickness (IVS, mm) were measured by manually tracing the endo-and epicardial borders in PSLAX M-mode images. Left ventricle volume in diastole and systole (LVVOLd, LVVOLs, respectively; µL) were calculated by the software as (7.0/(2.4+EDD))*EDD3, and (7.0/(2.4+ESD))*ESD3, respectively. The LV ejection fraction (LVEF, %) was measured as 100*(LVVOLd-LVVOLs)/LVVOLd. Stroke volume (SV, µL) was determined as LVIDd-LVIDs, and Cardiac Output (CO, mL/min) was estimated as SV*HR. To measure kinetics of endocardial wall contraction, M-mode velocity at the LVPW wall (MVel, mm/s) was determined by manually tracing the wall movement on the M-mode images.

Diastolic function was assessed by pulsed-wave Doppler (PWD) and tissue Doppler imaging (TDI) from apical 4 chamber views at the levels of the mitral valve (MV) and the septal annulus, respectively. Transmitral early (MV E, mm/s) and late atrial (MV A, mm/s) peak flow velocities, MV E/A ratio, and deceleration time (DecT, ms) of the E wave were determined. Isovolumic contraction time (IVCT, ms), systolic ejection time (SET, ms), and isovolumic relaxation time (IVRT, ms) were determined in the left ventricular cavity, where both mitral inflow and left ventricular outflow could be visualized. Myocardial performance index (MPI) was calculated as the Tei-index (IVRT+IVCT/LVET(SET)). The length of a cardiac cycle (CL; ms) was determined from R-R distance. Duration of systole (SystDur; ms) was considered as ET+IVCT, while duration of systole (DiastDur; ms) was calculated as CL-SystDur. The ratio of the DiastDur/SystDur was also calculated. The PW Doppler mode was also used to determine aortic flow (Ao) parameters from a modified suprasternal (aortic arch) view. Aortic velocity time integral (Ao VTI, mm), mean and peak velocity (Ao mean Vel, Ao peak Vel, mm/s), mean and peak pressure gradient (Ao mean Grad, Ao peak Grad, mmHg) were calculated by the software after automatically tracing the borders of the Doppler jet. Similarly, the flow parameters of the pulmonary artery (PA) were determined as PA velocity time integral (PA VTI, mm), mean and peak velocity (PA mean Vel, PA peak Vel, mm/s), mean and peak pressure gradient (PA mean Grad, PA peak Grad, mmHg; respectively). The pulmonary vein was visualized by using a modified PSLAX view [37]. Pulmonary vein systolic (PV S, mm/s) and diastolic (PV D, mm/s) deflections and PV atrial reversal (AR) peak velocity (PV Ar, mm/s) and duration (PV ARdur, ms) were measured. Tissue Doppler imaging (TDI) was performed at the septal annulus to evaluate peak tissue velocities at systole (s’, mm/s), and in early (e’ mm/s) and late (a’, mm/s) filling. The ratio of E/e’ was then determined. Three cardiac cycles were averaged for each parameter.

### 4.7. Strain Echocardiography

Strain echocardiographic analysis was carried out offline. According to the Lagrangian and Eulerian strain tensors of finite deformation theory, the strain (S) of a soft tissue in each direction is described as the difference between the segment’s initial length and its changed length, whereas strain rate (SR) is the rate of deformation change over a given time period. During the conventional echocardiographic imaging, high-frame rate (>200 fps) traces were recorded from both the parasternal long- and short axis views (PLAX and SAX, respectively), by a trained investigator. Images were evaluated in slow motion (1/8) before data storage, to ensure reproducibility. A single observer performed all strain analysis using the VevoLAB^®^ software equipped with the VevoStrain^®^ extension (ver. 5.6.0; FUJIFILM VisualSonics, Amsterdam, Netherlands). The software displays acquired videos in configurable slow-motion loops, allowing for suitable analysis even when the HR is high. Grayscale B-mode pictures were used to measure strain and strain rate parameters. Images were carefully selected to provide adequate visualization of myocardial borders; endo- and epicardial borders were traced semi-automatically, with manual adjustment where needed, to achieve proper quality tracking of each loop. Borders were traced from the mid-basal level. Longitudinal and radial strain parameters were generated from PLAX view, while circumferential strain parameters and fractional area change (FAC; %) were assessed from SAX view images, with papillary muscles excluded from tracing. Ventricular borders were defined, and the speckle-tracking system assigned circumjacent areas adjacent to the outlined chamber. The software algorithm estimated velocity (V; cm/s), displacement (D; mm), strain (S; %), and strain rate (SR; s-1) for each site on the designated line by following the defined border region (longitudinal and radial strain values were analyzed at the endocardial border). Speckle-tracking data were visualized in a color-coded map representing diastolic and systolic deformations, and reconstructed in a 3D image to provide a better spatio-temporal perception of wall motion. Time-to-peak analysis (TPk) was also performed to assess local wall motion synchronism, as TPk slope analysis delivers further information about regional motion and deformation development. For this method, the myocardium was divided into six anatomic segments, in agreement with the definition of the American Heart Association. Peak and average radial systolic velocity values (cm/s), global longitudinal strain (GLS; %), global circumferential strain (GCS; %), and fractional area change (FAC; %) after i.v. danicamtiv administration were analyzed and compared to baseline values of each animal [37,38,39,40,41].

### 4.8. Electrocardiogram

Three lead ECG recording was performed in parallel with the echocardiographic examinations. After anesthesia, animals were placed in a dorsal position and leads were positioned subcutaneously. LabChart Reader v8.1.14 software was used to evaluate ECG recordings. The following parameters were considered for the evaluation: heart rate, PQ interval, QRS duration, QT time, corrected QT interval, T-wave amplitude. For the evaluation, 6 consecutive cardiac cycles were averaged for each parameter, and data are presented as mean ± SEM.

### 4.9. Data Analysis and Statistics

Results were evaluated and graphs were created in the GraphPad Prism 9.0 software (GraphPad Software, San Diego, CA, USA). As for cellular measurements the number of experiments in each group varied between 9 and 14 from 6 different hearts. Background fluorescence intensity levels were obtained at the end of the measurements on a region without cardiomyocytes and manually subtracted the fluorescence intensities for background correction. Fourteen rats were measured before and after iv. treatment of da-nicamtiv. For all experiments, values were evaluated for normality (Kolmogorov–Smirnov normality test) and were then evaluated by paired t-tests, ordinary one-way ANOVA, or Kruskal–Wallis test with multiple comparisons as appropriate. Group descriptions are given as mean ± SEM values. Statistical significance was accepted at *p* < 0.05; exact *p* values are indicated in the results section (rounded to three decimal places).

## 5. Conclusions

Danicamtiv has the potential to augment LV systolic function in the absence of an increase in the amplitude of the intracellular Ca^2+^ transient. Although the administration of danicamtiv inherently affects LV diastolic function, this effect may be less prominent for danicamtiv than for OM. To some degree, this difference may relate to danicamtiv’s evoked augmentation in atrial contractility. Moreover, limitations in diastolic function appear as a class-effect during the application of direct myosin activators. Accordingly, considerable attention should be paid to the stratification the of LV diastolic function in patients treated with danicamtiv.

## Figures and Tables

**Figure 1 ijms-24-00446-f001:**
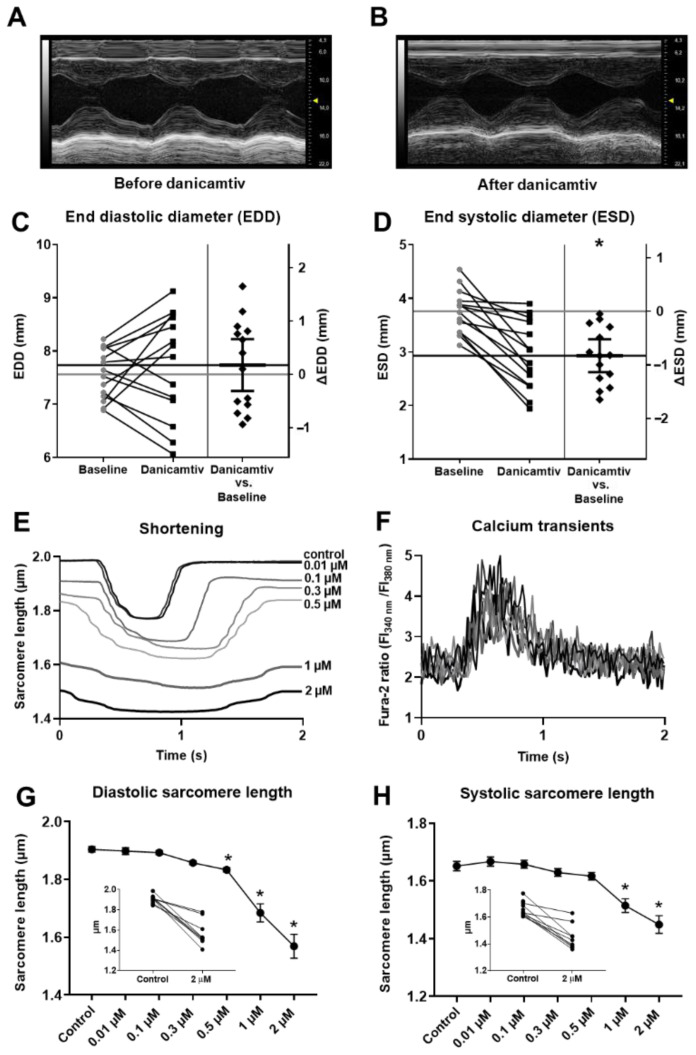
Danicamtiv distinctly affected LV dimensions and cardiomyocyte SL during systoles and diastoles. Representative M-mode echocardiographic images of rat LV before and after iv. danicamtiv treatment (**A**,**B**). Danicamtiv did not induce a significant decrease in the mean of EDD changes but it decreased significantly the mean of ESD (**C**,**D**). (Danicamtiv-dependent responses are illustrated for all animals on the left sides of panels (**C**,**D**). Differences between the values in the absence and presence of danicamtiv are shown on the right sides of panels (**C**,**D**) in individual experiments. A horizontal black dash in the middle illustrates the means of differences, and the continuous black lines at the same positions highlight the mean values after danicamtiv treatments. The grey line shows the mean of the baseline values, which coincides with 0 on the right Y axis. The vertical lines together with the two short black dashes on the right sides of panels (**C**,**D**) highlight 95% confidence intervals for the illustration of statistical significance (i.e., when it does not transect 0 level for the differences (ΔEDD or ΔESD)). Representative traces of SL changes during cardiomyocyte contractions and Ca^2+^ transients upon danicamtiv exposures (**E**,**F**). Different shades of gray refer to the applied drug concentrations, as indicated (**E**,**F**). Danicamtiv prolonged the duration of cardiomyocyte contractions, but did not affect Ca^2+^ transient durations. Significant reductions both in diastolic (**G**) and systolic (**H**) SL were observed in isolated left ventricular canine cardiomyocytes upon danicamtiv administration. Insets show values of SL of individual cardiomyocytes in control and in 2 µM danicamtiv. Symbols represent the means and standard errors of the mean (SEM). Significant differences from the baseline values are indicated by asterisks (* = *p* < 0.05; exact *p* values are indicated in the text).

**Figure 2 ijms-24-00446-f002:**
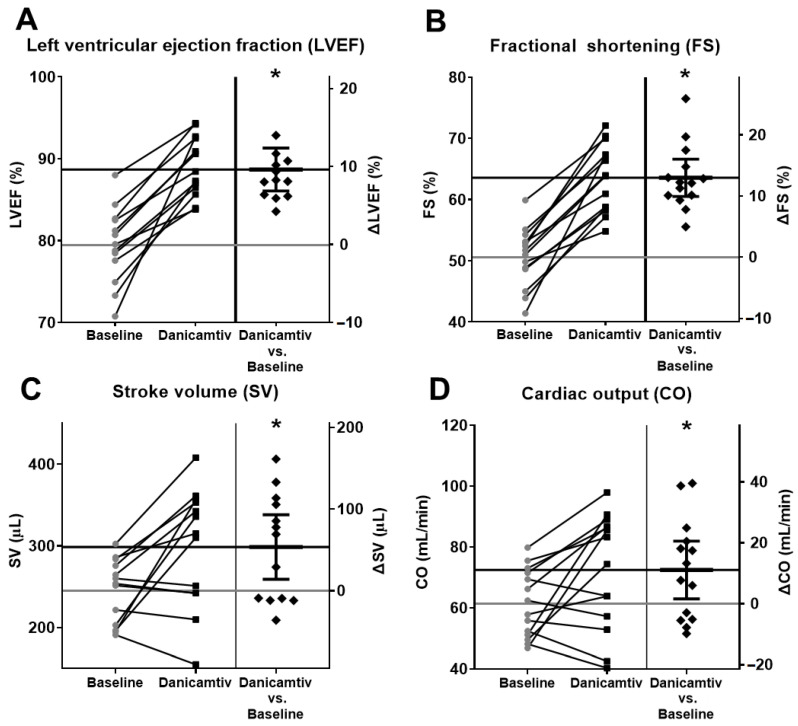
Danicamtiv augmented left ventricular systolic function. Danicamtiv improved LV systolic contractile parameters: LV ejection fraction (EF) (**A**), fractional shortening (FS) (**B**), and stroke volume (SV) (**C**). Cardiac output also increased upon danicamtiv administration (**D**). (The construction of panels is identical with those in Figure 1C,D). * = *p*  <  0.05.

**Figure 3 ijms-24-00446-f003:**
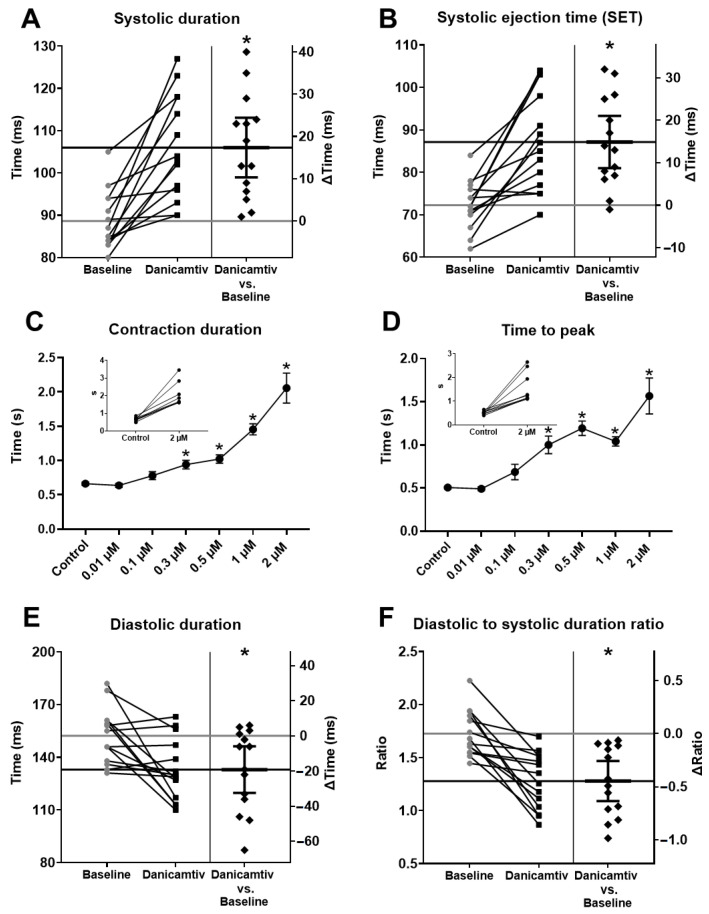
Danicamtiv increased the durations of systoles and decreased the durations of diastoles. Danicamtiv prolonged both systolic durations and systolic ejection time (**A**,**B**). When danicamtiv was added to the tissue chamber in different concentrations (0.01 µM to 2 µM), a progressive increase in contraction duration and time to peak interval could be observed (**C**,**D**). The results of individual experiments are shown in the insets. The symbols represent the mean and standard error of the mean. Significant differences from the baseline values upon application of danicamtiv are indicated by asterisks. Danicamtiv decreased diastolic duration (**E**) with a consequent decline in the ratio of diastolic to systolic durations (**F**). (The construction of panels is identical with the corresponding ones in Figure 1). * = *p*  <  0.05.

**Figure 4 ijms-24-00446-f004:**
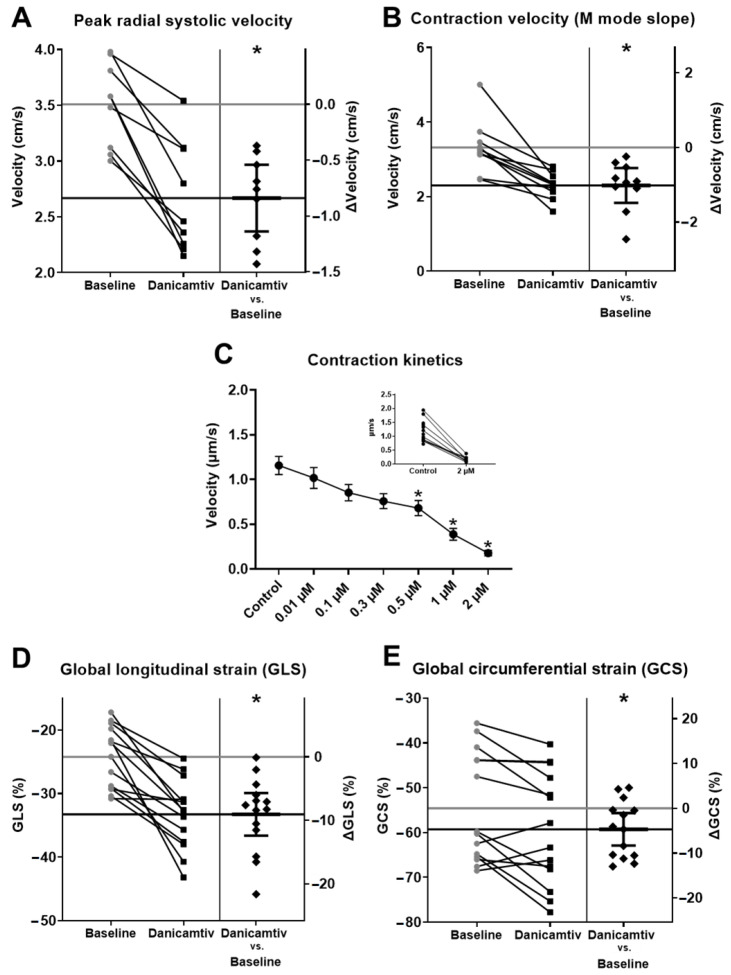
Danicamtiv decreased LV and cardiomyocyte contraction kinetics. The LV peak radial systolic velocity and contraction velocity (M-mode slope) decreased significantly upon danicamtiv administration (**A**,**B**). Cardiomyocyte contraction kinetics were also slowed down by danicamtiv (**C**). Global longitudinal and circumferential strain were decreased significantly by danicamtiv (**D**,**E**). (The construction of panels is identical with the corresponding ones in Figure 1). * = *p*  <  0.05.

**Figure 5 ijms-24-00446-f005:**
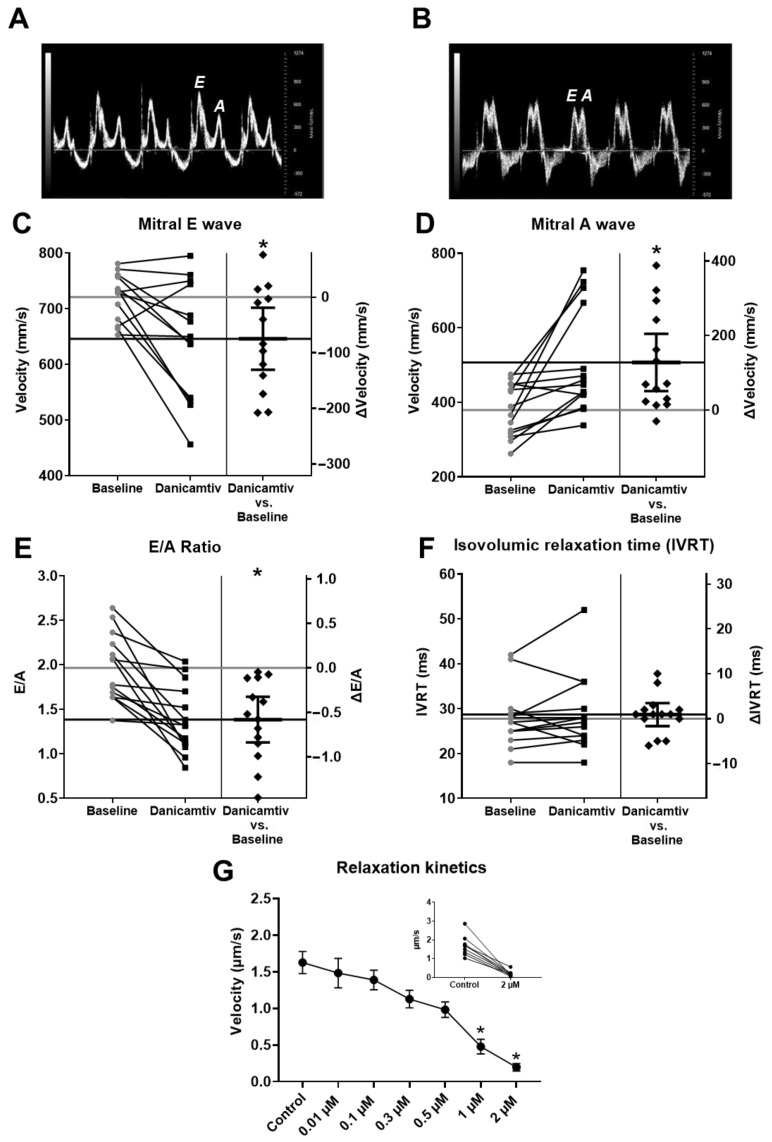
Danicamtiv impaired LV diastolic function and relaxation kinetics. Pulsed-wave Doppler imaging at the mitral valve was used to determine the early/atrial inflow ratio (E/A) (representative individual recordings in panel (**A**) for control and presence of danicamtiv in panel (**B**)). Danicamtiv negatively affected LV diastolic filling in the rat (**C**–**E**) Consistent with the echocardiographic measurements, danicamtiv significantly decreased relaxation kinetics in freshly isolated canine LV cardiomyocytes (**F**,**G**). The IVRT remained unaltered. Cardiomyocytes were treated with danicamtiv (0.01, 0.1, 0.3, 0.5, 1, and 2 µM) during the measurements. (The construction of panels is identical with the corresponding ones in Figure 1). * = *p*  <  0.05.

**Figure 6 ijms-24-00446-f006:**
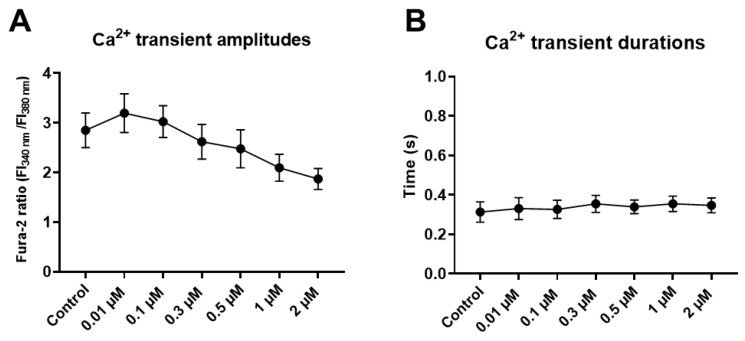
Danicamtiv did not alter intracellular Ca^2+^ transients in canine cardiomyocytes. Ca^2+^ transients were recorded in parallel with contractile parameters. No significant differences were observed in the Ca^2+^ transient amplitudes (**A**) and durations (**B**) with danicamtiv.

**Table 1 ijms-24-00446-t001:** Echocardiographic and electrocardiographic parameters measured at baseline and following danicamtiv administration in rats.

Parameters	Baseline (n = 14)	Danicamtiv (n = 14)	*p* Value (Baseline vs. Danicamtiv)
*Echocardiography*			
LV volume—systole (μL)	61.53 ± 4.23	35.7 ± 4.77	<0.001
LV volume—diastole (μL)	305.4 ± 11.2	326.1 ± 23.9	0.327
LV anterior wall thickness—systole (mm)	3.17 ± 0.09	3.49 ± 0.14	0.034
LV anterior wall thickness—diastole (mm)	1.76 ± 0.22	1.65 ± 0.26	0.171
LV posterior wall thickness—systole (mm)	3.24 ± 0.12	3.69 ± 0.09	0.003
LV posterior wall thickness—diastole (mm)	1.93 ± 0.08	1.92 ± 0.08	0.867
Aorta peak systolic velocity (mm/s)	899.3 ± 35.7	936.7 ± 41.2	0.276
Aorta mean systolic velocity (mm/s)	547.8 ± 29.8	511.1 ± 35.4	0.213
Aorta peak pressure gradient (mmHg)	3.30 ± 0.26	3.59 ± 0.31	0.248
Aorta mean pressure gradient (mmHg)	1.30 ± 0.15	1.11 ± 0.17	0.178
Pulmonic vein atrial reverse flow (mm/s)	136.7 ± 11.4	201.1 ± 24.9	0.008
Pulmonic vein atrial reverse flow duration (ms)	20.27 ± 0.77	28.10 ± 2.24	0.004
Mitral E wave deceleration time (ms)	57.57 ± 2.53	51.09 ± 2.63	0.134
Mitral valve septal e’ (mm/s)	40.47 ± 2.44	36.36 ± 2.36	0.051
Mitral valve septal a’ (mm/s)	35.21 ± 1.74	42.24 ± 2.27	0.070
Mitral valve septal e’/a’	1.21 ± 0.11	0.874 ± 0.10	0.014
Tei index	0.60 ± 0.04	0.50 ± 0.03	<0.001
E/e’ ratio	18.79 ± 1.31	19.53 ± 1.69	0.549
Isovolumic contraction time (ms)	14.50 ± 0.53	14.50 ± 0.91	0.999
Isovolumic relaxation time (ms)	27.79 ± 1.77	28.71 ± 2.22	0.450
*Electrocardiography*			
Heart rate (bpm)	251 ± 15	249 ± 14	0.715
PQ interval (s)	0.048 ± 0.004	0.051 ± 0.004	0.245
QRS duration (s)	0.017 ± 0.003	0.017 ± 0.002	0.869
QT interval—corrected to heart rate (s)	0.042 ± 0.002	0.042 ± 0.001	0.577
T wave amplitude (mV)	0.13 ± 0.02	0.13 ± 0.03	0.477

## Data Availability

The data analyzed and presented in this study are available from the corresponding author on request.

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
