# Peer review of "The Novel Cardiac Myosin Activator Danicamtiv Improves Cardiac Systolic Function at the Expense of Diastolic Dysfunction In Vitro and In Vivo: Implications for Clinical Applications"

_ijms, 2022, doi:10.3390/ijms24010446_

Round 1

Reviewer 1 Report

The authors have provided clear evidence that the novel cardiac myosin activator danicamtiv improves cardiac systolic function in expense of diastolic dysfunction. Thereby the effects of danicamtiv were studied both in vitro and in vivo and implications for possible clinical applications given.

The reported data are novel and very clearly presented. The utilized methods are more than sufficient. From my side, there is only a minor issue and a suggestion which should be addressed.

Minor issue:

Please summarize in one paragraph in the Discussion the most important differences of danicamtiv vs. omecamtiv mercabil (OM). This would be helpful for the readers.

Minor point:

For the print version it would be helpful if the font size for the labeling of the Figures (where possible) would be slightly increased (e.g. “Baseline”; “Danicamtiv”; “Danicamtiv vs. Baseline” –  Fig. 2, 3, 4 and 5).

Author Response

Responses to the reviewer’s comments

We are grateful for the editor and the reviewers for their efforts they invested and for providing rapid review on our paper (Manuscript ID: ijms-2116363). We also thank the editorial team for the encouraging decision and for asking minor revisions of our manuscript. We dealt with all comments carefully. Kindly, find our point-by-point responses below:

Reviewer #1

Comments and Suggestions for Authors

The authors have provided clear evidence that the novel cardiac myosin activator danicamtiv improves cardiac systolic function in expense of diastolic dysfunction. Thereby the effects of danicamtiv were studied both in vitro and in vivo and implications for possible clinical applications given. The reported data are novel and very clearly presented. The utilized methods are more than sufficient. From my side, there is only a minor issue and a suggestion which should be addressed.

Response: We thank the reviewer for his/her thoughtful comments and kind words. During the revision we incorporated your suggestions. The manuscript improved significantly upon the proposed revision.

Minor issue:

Please summarize in one paragraph in the Discussion the most important differences of danicamtiv vs. omecamtiv mercabil (OM). This would be helpful for the readers.

Response: Thank you for this important remark. We accepted you request and added a short paragraph in which now we discuss the main similarities and differences between the two drug candidates.

Change in text: Taken together, cardiac effects of OM and danicamtiv are largely similar: both compounds effectively enhance LV systolic function, although they can also limit LV diastolic function. The data presented here implicate advantages for danicamtiv over OM, since danicamtiv did not affect IVRT or IVCT. Moreover, left atrial contractile function was enhanced at a relatively low danicamtiv concentration. To our knowledge, similar effects have not been reported for OM yet. (new paragraph at the end of Discussion)

Minor point:

For the print version it would be helpful if the font size for the labeling of the Figures (where possible) would be slightly increased (e.g. “Baseline”; “Danicamtiv”; “Danicamtiv vs. Baseline” – Fig. 2, 3, 4 and 5).

Response: Thank you for this comment. Labelling and font sizes have been updated in most figures to facilitate reading and understanding as suggested.

Reviewer 2 Report

First of all, I want to congratulate the authors for their interesting study and the well-written manuscript!

The authors performed a mixed in-vitro (insulated canine cardiomyocytes) and in-vivo (rat hearts) study to investigate the effect of the myosin activator danicamtiv. They found significant positive effects on myocardial function (i.e. increased ventricular ejection fraction, change of systolic/diastolic ratio), showing the potential of this drug in heart failure treatment.

After careful revision I suggest to revise some (minor) points of the submitted manuscript:

- Unfortunately, no heart failure model was used in this study. It would have been interesting to see the effect of the myosin activator in this scenario. I recommend to discuss this in the manuscript and/or to add it as a limitation.

- Introduction, p. 2, l.64-67: please add a reference

- section 2.2, p.3, l. 122: the abbreviation of ejection fraction was already introduced above (p. 2, l.66)

- sections 4.1.1 and 4.3: It is not clear, why the authors mention mongrel dogs. To my understanding, isolated canine cardiomyocytes and (anaethesized) rats were used in the study. Please clarify!

Author Response

Responses to the reviewer’s comments

We are grateful for the editor and the reviewers for their efforts they invested and for providing rapid review on our paper (Manuscript ID: ijms-2116363). We also thank the editorial team for the encouraging decision and for asking minor revisions of our manuscript. We dealt with all comments carefully. Kindly, find our point-by-point responses below:

Reviewer #2:

Comments and Suggestions for Authors

First of all, I want to congratulate the authors for their interesting study and the well-written manuscript!

The authors performed a mixed in-vitro (insulated canine cardiomyocytes) and in-vivo (rat  hearts) study to investigate the effect of the myosin activator danicamtiv. They found significant positive effects on myocardial function (i.e. increased ventricular ejection fraction, change of systolic/diastolic ratio), showing the potential of this drug in heart failure treatment.

Response: We would like to express our gratitude for your supportive comments.

After careful revision I suggest to revise some (minor) points of the submitted manuscript:

- Unfortunately, no heart failure model was used in this study. It would have been interesting to see the effect of the myosin activator in this scenario. I recommend to discuss this in the manuscript and/or to add it as a limitation.

Response: Thank you for this important remark. As you kindly requested, we added a short paragraph to discuss this issue as a limitation of the study at the end of the Discussion section of the revised manuscript.

Change in text: It is to be acknowledged, that extrapolation of our results to the failing heart can be limited by the fact that our investigations were carried out on cardiac preparations of healthy animals. Nevertheless, our results are in accord with those performed in a canine model of heart failure and in a phase2a trial on danicamtiv. (new paragraph at the end of Discussion)

 - Introduction, p. 2, l.64-67: please add a reference

Response: New reference was added: Shen YT, Malik FI, Zhao X, et al. Improvement of cardiac function by a cardiac myosin activator in conscious dogs with systolic heart failure. Circ Heart Fail. 2010;3(4):522-527. doi:10.1161/CIRCHEARTFAILURE.109.930321

  - section 2.2, p.3, l. 122: the abbreviation of ejection fraction was already introduced above  (p. 2, l.66)

Response: Thank you, the correction was made.

- sections 4.1.1 and 4.3: It is not clear, why the authors mention mongrel dogs. To my understanding, isolated canine cardiomyocytes and (anaethesized) rats were used in the study. Please clarify!

Response: Thank you for this remark. To facilitate better understanding, we rephrased two sentences in the Methods section. Isolated canine cardiomyocytes were used for the current study for in vitro measurements which were originated from mongrel dogs, and rats were used for the in vivo experiments.

Changes in text: 

„Cardiomyocytes with intact membranes were isolated from adult mongrel dogs” (Methods)

„Canine cardiomyocytes obtained from adult mongrel dogs were studied since their physiological properties are very similar to those of humans.” (Methods)